# Domain Adaptation Speech-to-Text for Low-Resource European Portuguese Using Deep Learning

Eduardo Medeiros [1,*], Leonel Corado [1], Luís Rato [1,2,*], Paulo Quaresma [1,2] and Pedro Salgueiro [1,2]

1   Escola de Ciências e Tecnologia, Universidade de Évora, 7000-671 Évora, Portugal;
    leonel.corado@uevora.pt (L.C.); pq@uevora.pt (P.Q.); pds@uevora.pt (P.S.)
2   Centro ALGORITMI, Vista Lab, Universidade de Évora, 7000-671 Évora, Portugal
*   Correspondence: efarofia@uevora.pt (E.M.); lmr@uevora.pt (L.R.)

**Abstract:** Automatic speech recognition (ASR), commonly known as speech-to-text, is the process of transcribing audio recordings into text, i.e., transforming speech into the respective sequence of words. This paper presents a deep learning ASR system optimization and evaluation for the European Portuguese language. We present a pipeline composed of several stages for data acquisition, analysis, pre-processing, model creation, and evaluation. A transfer learning approach is proposed considering an English language-optimized model as starting point; a target composed of European Portuguese; and the contribution to the transfer process by a source from a different domain consisting of a multiple-variant Portuguese language dataset, essentially composed of Brazilian Portuguese. A domain adaptation was investigated between European Portuguese and mixed (mostly Brazilian) Portuguese. The proposed optimization evaluation used the NVIDIA NeMo framework implementing the QuartzNet15×5 architecture based on 1D time-channel separable convolutions. Following this transfer learning data-centric approach, the model was optimized, achieving a state-of-the-art word error rate (WER) of 0.0503.

**Keywords:** machine learning; deep learning; deep neural networks; speech-to-text; automatic speech recognition; NVIDIA NeMo; GPUs; data-centric; Portuguese language





## 1. Introduction

Automatic speech recognition (ASR), commonly known as speech-to-text, is the process of converting sounds into text, i.e., transforming speech into the respective sequence of words. In today's world, most of the writing process is done on computers, tablets, or smartphones. Although typing is predominant, dictation has shown to be faster [1], which opens the opportunity for ASR to become the primary way for writing text. Despite this scenario, some challenges still persist in the ASR field, especially regarding its efficiency in less favorable conditions, e.g., noisy environments, or its application in low-resource languages.

In recent years, the field of artificial intelligence (AI) and related fields, such as machine learning (ML), have seen major growth in popularity and in technological advancements. Deep learning (DL), a subset of ML, makes use of deep neural networks (DNNs), a special type of artificial neural network (ANN), with a large number of layers. This large amount of layers allows features to be extracted from the raw input data without the need for any pre-processing [2]. Alongside a very rapid growth of available data, DNNs have started to be used in a variety of fields, such as computer vision, natural language processing, and speech recognition.

The Portuguese language is one of the most spoken languages in the world and it is divided into different variants, such as Brazilian and European Portuguese [3]. Among other variants, European Portuguese represents a small fraction of Portuguese speakers, and hence has fewer available resources and research. Developing deep neural models requires large datasets, which are scarce or not available for the European variant of the

Portuguese language. Thus, transfer learning can be used as a means of creating ASR systems for languages with limited amounts of data. Such can be achieved by transferring the knowledge from models developed for languages with a larger amount of accessible data, such as the English language. The pre-trained models are then tuned for languages with less available data, such as the Portuguese language [4]. The aforementioned state-of-the-art methodology and the scarcity of resources for European Portuguese are the main motivation for the current work. This work aims to develop an ASR model using deep learning techniques for the Portuguese language. Using NVIDIA NeMo [5], we optimized the QuartzNet15×5 [6] architecture to develop a model using a data-centric approach which includes data pre-processing; transfer learning from English language models; and a domain adaptation evaluation, considering a strictly European Portuguese target and mixed (mostly Brazilian) Portuguese sources, which yielded a state-of-the-art word error rate (WER) performance.

## 2. Literature Review

This section introduces the main topics of the current work: deep learning and its applications (Section 2.1) and ASR for the Portuguese language (Section 2.2).

### 2.1. Deep Learning in ASR

Recent years have seen a growth in knowledge about the application of ANNs in a variety of research areas, including ASR, computer vision (image processing and recognition), and natural language processing. End-to-end (E2E) methods for ASR systems have increased in popularity due to these developments and the exponential growth in the amount of data available [7].

Approaches to automatic speech recognition systems are largely based on four types of deep neural networks: convolutional neural networks (CNNs) [8]; recurrent neural networks (RNNs) [9]; time-delay neural networks (TDNNs) [10]; and most recently, transformers [11]. These architectures can employ E2E mechanisms such as attention-based encoders and decoders (AED), recurrent neural network transducer (RNN-T), and connectionist temporal classification (CTC) [7].

The CTC algorithm is commonly used in ASR, as well as handwriting and other problems, the output of which consists of a sequence. When speech data only contain the audio and the transcript, there is no direct correlation between them. The correct alignment is possible by applying the CTC loss function on a DNN. This decoding process of aligning speech to words makes use of a tokenizer that contains the vocabulary (or alphabet) of the target audio language.

The Jasper model architecture is a good example of this [12]. It uses a TDNN architecture trained with CTC loss. This model achieved best performance of 2.95% WER on a test clean LibriSpeech dataset and improved over various domains compared to other state-of-the-art models.

Many applications in ASR infer pre-trained models from APIs due to their performance for the English language, as shown in Table 1.

**Table 1.** API architecture and respective training hours.

| Architecture | Train Hours |
| --- | --- |
| Facebook wav2vec 2.0 | 1041 |
| Google LAS | 2025 |
| Microsoft ASR System | 12,500 |

These models can also be fine-tuned to other languages if given new vocabulary and speech data. This process is also known as transfer learning (TL).

These types of DNNs require large quantities of data to be trained and achieve good performance. When the available data are scarce, the transfer learning technique can be

applied to these networks to improve their performance. Transferring knowledge on ANN-based systems is the equivalent of reusing layers from previously trained models. This is accomplished by using previously calculated weights to initialize specified layers of new models, followed by training the remaining layers. The reused layers can either be fixed, in which case the pre-calculated weights will not be updated, or flexible, in which case the pre-calculated weights will be able to be updated according to the new data [13]. The weights of the remaining layers are randomly initialized as in a normal ANN.

Sampaio et al. [14] evaluated the APIs from Table 1 using two collaborative and public Portuguese datasets, Mozilla Common Voice (https://commonvoice.mozilla.org/pt) (accessed on 11 November 2021) and the Voxforge (http://www.voxforge.org/pt) (accessed on 11 November 2021). Each had a domain, the former being common words and utterances, while the latter was audiobooks. The result of each API over each dataset can be seen in Table 2.

**Table 2.** APIs results on Mozilla Common Voice (MCV) Corpus and Voxforge Corpus datasets.

| Architecture | MCV WER | Voxfoge WER |
|:---:|:---:|:---:|
| Facebook wav2vec 2.0 | 12.29% | 11.44% |
| Google LAS | 12.58% | 10.49% |
| Microsoft ASR System | 9.56% | 7.25% |

The model can adapt to new languages or domains if given enough training data for it to transfer to new vocabularies. Transferring knowledge from a high-resource language to a low-resource language, such as Portuguese, has been shown to improve the low-resource ASR model [15,16].

### 2.2. ASR in the Portuguese Language

The Portuguese language is one of the most spoken in the world, not due to the size of the Portuguese population (10 million), which gives the language its name, but thanks to countries with a much larger number of inhabitants, such as Brazil (214 million), Angola (34 million), and Mozambique (32 million). Despite speaking the same language, the speech varies from country to country and even from region to region, not only in accent but also in vocabulary. The goal of this work is to use European Portuguese (EP), i.e., from Portugal, to develop an ASR system.

As already mentioned, EP has fewer speakers than other variants such as Brazilian Portuguese (BP). However, some research has already been developed in the field of ASR with the goal of transcribing EP speech. Pellegrini et al. [17] and Hämäläinen et al. [18] aimed to transcribe speech from elder and young people since in these age groups people have more difficulties expressing themselves. The goal was to improve the understatement of their speech through the use of ASR systems. Other research aimed to create a speech recognizer for EP based on a corpus obtained from broadcast news and newspapers. The AUDIMUS.media [19] speech recognizer is a hybrid system composed of an ANN, a multilayer perceptron (MLP), which classifies phones according to features extracted by a Perceptual Linear Prediction (PLP), a log-RelAtiveSpecTrAl (Log-RASTA), and Modulation Spectrogram (MSG). These components are then combined and used in an HMM for temporal modeling [20].

In variants with a larger amount of speakers, such as Brazilian Portuguese, there is also a lack of results related to the development of ASR systems. This shortage is mostly due to the lack of data quantity, quality, or detail in public datasets, or lack of public datasets to begin with, though they are desperately needed, especially when creating models based on DNNs.

Lima et al. [21] provided a list of 24 Portuguese language datasets alongside some of the available features, such as size, quality, rate, amount of speakers, speaker's age, and availability (public or private). Of the twenty-four datasets, only six are public, which leads Lima et al. to state that the amount of datasets available is acceptable to build ASR systems for the Portuguese language. Lima et al. also conclude that the types of data are diverse (noisy, wide age range, medical, commands), but the overall quantity, quality, and standardization are poor.

Nevertheless, some research has shown that it is possible to create models for ASR systems for the Portuguese language using reduced amounts of data, as little as 1 h, and still achieve considerable results in word error rate (WER), as high as 34% [22]. Works regarding ASR systems for Portuguese using DNNs worth mentioning include Gris et al. [23], who make use of Wav2vec 2.0 and pre-trained models in other languages (which are then fine-tuned to BP) and achieve an average WER of 12.4% on seven datasets; and Quintanilha et al. [24,25], who make use of four datasets (three of which are open) and use models based on DeepSpeech 2 [26] with convolutional and bidirectional recurrent layers, making it possible to achieve values of 25.45% WER. Additional work is available regarding ASR systems for the Portuguese language. The app TipTopTalk! [27] by Tejedor-García et al. uses Google's ASR systems to implement a pronunciation training application for various languages including both European and Brazilian variants of Portuguese.

## 3. Data

The performance of ASR systems does not rely only on the type of algorithms used; it also depends on the quality and quantity of the available data. Some languages, such as English, are widely present in the majority of the ASR applications of today's world, e.g., smart assistants, with large amounts of research in the area of ASR, e.g., [28]. To build an ASR system, data may be acquired through private entities, open-access datasets, crowdsourcing, or by creating the dataset from scratch, e.g., collecting audiobooks and their transcriptions. Although scarce, some sources of audio and their transcriptions are available in Portuguese. Nevertheless, the quantity and quality of publicly available datasets are not usually good enough to create high-performance ASR systems. These sources usually lack audio or transcription quality, data quantity, and structure standardization. The following sections describe the data sources used in this project. Section 3.1 describes the LibriSpeech dataset, used to evaluate the effectiveness of the transfer learning technique, and the Multilangual LibriSpeech dataset. Section 3.2 describes the SpeechDat dataset. These datasets were used in the current work to perform the data-driven experiments focused on transfer learning and domain adaptation.

### 3.1. LibriSpeech and Multilingual LibriSpeech

As the amount of content available online increases, more data can be collected for research purposes. The LibriSpeech (https://www.openslr.org/12) (accessed on 12 December 2021) dataset is built with data from the LibriVox (https://librivox.org) (accessed on 6 January 2022) audiobook catalogue. Composed of 1000 h of audio recordings of English speech, with a sampling rate of 16 kHz, LibriSpeech was created with the purpose of being used to build and test ASR systems. In the work presented in this paper, the training set of 100 h of "clean" speech of the LibriSpeech (https://www.openslr.org/resources/12/train-clean-100.tar.gz) (accessed on 12 December 2021) dataset was used as a control dataset for the English domain [29]. This control was meant to verify the performance of the transfer learning process, i.e., to verify if the models pre-trained in English were properly learning the Portuguese data. The 100 h set contains transcripts with a total of 990,101 words, from which 33,798 are unique words.

Multilingual LibriSpeech (MLS) is an extension of LibriSpeech which increases the amount of English speech to 44,500 h of audio recordings and adds 6000 h of audio recordings from seven other languages, including Portuguese [30].

MLS provides the dataset split into three subsets—train, development, and test—in which there is no speaker overlap. For the last two sets, it is also guaranteed that the gender and duration of the speaker are balanced. Audio recordings are also ensured to unambiguously contain only one speaker. With respect to Portuguese, MLS contains a large number of hours ($\simeq$167 h of Brazilian Portuguese and $\simeq$1 h of European Portuguese audio recordings). These hours correspond to a total of 1,321,326 words, of which 77,292 are unique, with an average of 33.68 words per transcription.

### 3.2. SpeechDat

The SpeechDat European Project (https://cordis.europa.eu/project/id/LE24001) (accessed on 17 July 2022) was developed between 1996 and 1998 with the goal of providing speech resources to stimulate research and development of automated services, such as speech recognizers. The Portuguese database was collected by Portugal Telecom, now named Altice Portugal, in collaboration with INESC and INESCTEL.

The current work was developed as a collaboration between Universidade de Évora and Altice Labs (Altice Labs is part of Altice Portugal), which enabled the current work to study ASR in Portuguese using the SpeechDat Portuguese dataset. SpeechDat audio recordings consist of 4027 Portugal Telecom employees following prompt sheets. This dataset has a good geographical distribution and a good representation of regional European Portuguese accents of native and non-native, as shown in Table 3.

**Table 3.** Distribution of native and non-native speakers.

| Region | Number of Speakers | Number of Speakers (%) |
|---|---|---|
| Entre-Douro-e-Minho | 1048 | 26 |
| Transmontano | 202 | 5 |
| Beira-Litoral | 537 | 13 |
| Beira-Alta | 227 | 6 |
| Beira-Baixa | 97 | 2 |
| Estremadura | 858 | 21 |
| Ribatejo | 151 | 4 |
| Alentejo | 325 | 8 |
| Algarve | 148 | 4 |
| Azores | 163 | 4 |
| Madeira | 95 | 2 |
| Portuguese-speaking countries | 116 | 3 |
| Other countries | 29 | 1 |
| Empty field | 31 | 1 |

According to the documentation, speakers from "Portuguese-speaking countries" (which includes Macau and India, since there are speakers from these regions where there were Portuguese colonies and there exist Portuguese-speaking communities) have been living in Portugal for many years, so their dialectal differences are not so easily distinguishable. The speakers born in "other countries" are relatives of Portuguese emigrants.

Table 4 displays the distribution of the speakers' sexes and age groups. The majority of the speakers belong to the 16–30 and 31–45 age groups. The female sex ($\simeq$54%) is slightly more represented than the male sex ($\simeq$46%).

**Table 4.** Distribution of speakers over age groups and sexes.

| Age Groups | Male Speakers | Female Speakers | Percentage of Total |
|---|---|---|---|
| <16 | 93 | 148 | 6.0 |
| 16-30 | 581 | 823 | 34.9 |
| 31-45 | 742 | 790 | 38.0 |
| 46-60 | 376 | 335 | 17.7 |
| >60 | 69 | 70 | 3.4 |

The audio recordings, consisting of small sentences, were encoded using A-LAW [31] (an algorithm used for encoding audio signals, in particular voice encoding) with a sampling rate of 8 kHz 8 bit, which was accompanied by an ASCII label file containing the audio recording metadata.

Each label file contained an assessment code regarding the quality of the respective audio. The possible values for the code are the following:

- OK—clean audio and ready to be used;
- NOISE—audio with some background noise;
- GARBAGE—empty audio, missing transcriptions, only background noise, noise produced by others;
- OTHER—audio containing disfluencies, hesitations, stuttering, or unintelligible speech;
- NO_PTO—audio files without label file.

Table 5 presents the number of hours of audio recordings of each audio quality label and the total hours of audio recordings of the SpeechDat dataset. The 186 total hours of audio recordings represent a lexicon of approximately 15,000 different words, the smaller recordings being ≃0.5 s and the larger recordings being ≃20 s.

**Table 5.** Hours of audio recordings for each audio quality label.

| Quality Label | Hours |
| --- | --- |
| OK | 152.99 |
| NOISE | 30.82 |
| GARBAGE | 1.04 |
| OTHER | 0.34 |
| NO_PTO | 0.90 |
| TOTAL | 186.09 |

We can observe that the SpeechDat dataset is slightly larger than the Multilingual LibriSpeech, by ≃18 h.

## 4. Proposed System

The current Section presents the proposed system capable of transcribing Portuguese audio into the respective transcription using deep neural models. Such a system is achieved through the use of a pipeline and a framework that enables the development of deep neural models with the selected neural architecture.

### 4.1. Architecture

A pipeline was assembled to create models for transcribing Portuguese speech into the respective transcription using deep learning to facilitate connections between each necessary stage of this process. Figure 1 illustrates the pipeline built, whose stages are described below:

- Dataset Acquisition—explore and acquire new speech datasets in Portuguese (audio recording files and respective transcriptions);
- Content Analysis—dataset analysis, assessment of the dataset's initial structure, documentation, content quantity and quality, and audio file encodings;
- Pre-Processing—dataset restructuring, data pre-processing, and manifest creation;
- Model Creation—model creation with the pre-processed data;
- Results Evaluation—testing and evaluation of the created model.

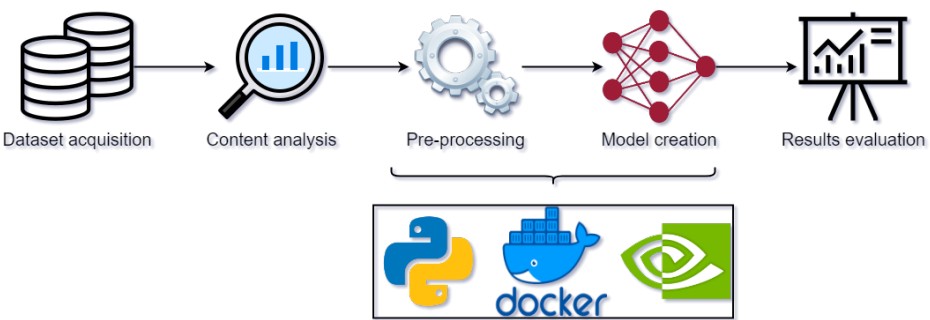

**Figure 1.** System pipeline and toolkit.

### 4.2. NVIDIA NeMo

State-of-the-art deep learning frameworks explore hardware features to their full potential. Multi-CPU, multi-GPU, and multi-node frameworks with high-speed inter-connectors allow algorithms to perform faster calculations and therefore bring down the training time of models.

NVIDIA NeMo (https://github.com/NVIDIA/NeMo) (accessed on 28 October 2021) [5] is a framework developed by NVIDIA built on top of the PyTorch and the PyTorch Lightning frameworks and is meant "(…) for building, training, and fine-tuning GPU-accelerated speech and natural language understanding (NLU) models with a simple Python interface." (https://developer.nvidia.com/nvidia-nemo). NeMo provides separate collections for automatic speech recognition, natural language processing, and text-to-speech models. Each collection consists of prebuilt modules that include everything needed to train new models. Every module can easily be customized, extended, and composed to create new conversational AI model architectures (https://docs.nvidia.com/deeplearning/nemo/user-guide/docs/en/stable/starthere/intro.html) (accessed on 15 February 2022).

Given the infrastructure in which the present work was developed and the features made available by NeMo, NeMo was the selected framework to develop the current work models.

### 4.3. Model Architecture

Regarding NeMo's automatic speech recognition collection, QuartzNet [6] and Jasper [12] are two end-to-end model architectures worth mentioning due to their role in the current work.

Jasper is a block architecture designed to ease fast GPU inference. The block architecture is composed of one convolutional pre-processing block, B × R blocks (where B represents the number of blocks and R the number of sub-blocks), and three convolutional post-processing blocks. Each block input is connected to the last sub-block using a residual connection. Each of the R sub-blocks applies a set of four operations: a 1D convolution, batch normalization, ReLU, and dropout. Models with this architecture are trained using the CTC loss function. Jasper models achieve state-of-the-art results of 2.95% when using a beam-search decoder and an external language model.

QuartzNet derives from Jasper's B × R block architecture and has the goal of achieving state-of-the-art results while using models with fewer parameters. QuartzNet performs similar operations, the only difference being the replacement of the 1D convolution with a 1D time-channel separable convolution (compared in Figure 2). The 1D time-channel separable convolution is an implementation of depthwise separable convolutions which can be separated into two other convolutions, a 1D depthwise convolution and a pointwise convolution (see Figure 3). The 1D depthwise convolutional layer performs convolutions across time, and a pointwise convolutional layer performs a 1 × 1 convolution across features/channels [6,32], which enables the QuartzNet architecture to achieve state-of-the-art results while drastically reducing the number of parameters.

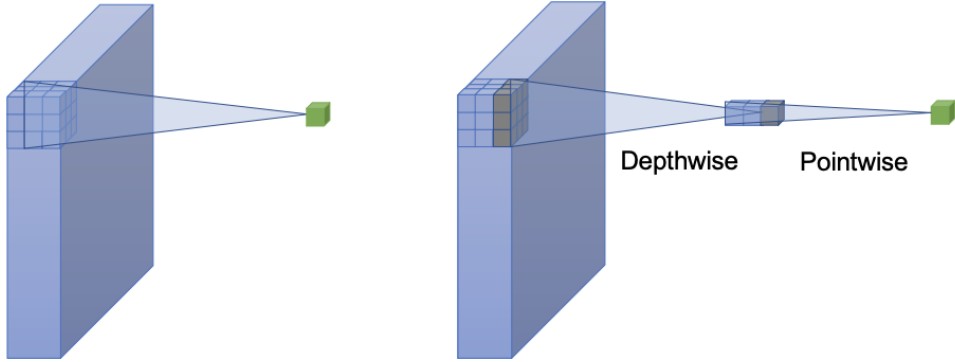

**Figure 2.** Standard convolution (**left**) and depthwise separable convolution (**right**) [33].

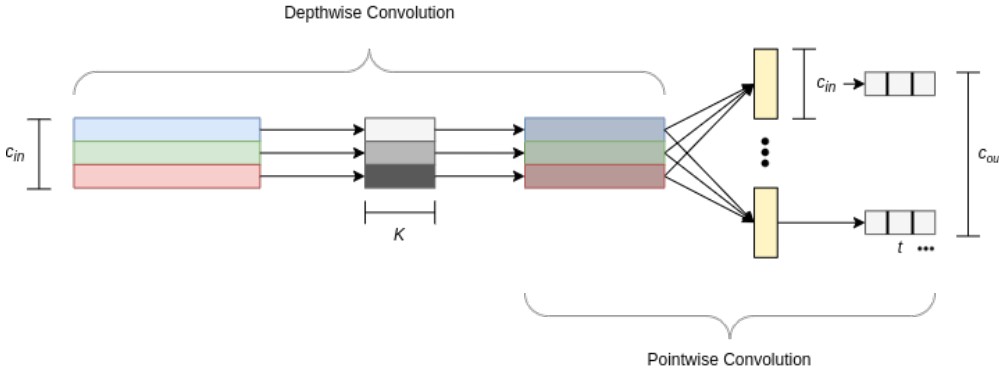

**Figure 3.** One-dimensional time-channel separable convolution.

### 4.4. Data Pre-Processing

The SpeechDat dataset, composed of ≃186 h of audio recording collected in controlled environments, was divided into two "versions", defined in Definitions 1 and 2.

**Definition 1** (SpeechDat with noise-describing tokens). *The data pre-processing carried out in this iteration of the dataset excluded audio recordings labelled as GARBAGE or OTHER, with transcriptions being ⁂, empty, or containing ~ or ∗.*

*The remaining audio recordings were re-encoded from A-LAW into WAV and re-sampled from 8 kHz to 16 kHz.*

**Definition 2** (SpeechDat without noise-describing tokens). *In addition to Definition 1, this iteration of the dataset had the noise-describing tokens [sta], [spk], [fil], and [int] removed from the transcriptions.*

In Table 6 are shown the splits used for each dataset in the present work. The SpeechDat dataset was divided into 70% (107,158 instances) for training/transfer, 15% (22,960 instances) for validation, and 15% (22,961 instances) for testing. The MLS dataset was already split having 37,498 instances for training, 826 for validation, and 906 for testing.

**Table 6.** Number of instances of each subset of the MLS and SpeechDat datasets.

| Subset | MLS | SpeechDat |
|---|---|---|
| Train | 37,498 | 107,158 |
| Validation | 826 | 22,960 |
| Test | 906 | 22,961 |

## 5. Experiments and Results

### 5.1. Hardware Infrastructure

The following experiments were run in the Vision Supercomputer (https://vision.uevora.pt/) (accessed on 23 January 2022). Vision is an HPC cluster made of two compute nodes, interconnected with 8 × 200 Gb/s HDR InfiniBand links for parallel processing. The compute nodes are NVIDIA DGX A100 systems, with the following characteristics:

- GPUS: 8×NVIDIA A100 40 GB Tensor Core GPUs;
- GPU Memory: 320 GB total;
- CPU: Dual AMD Rome 7742, 128 cores total;
- Networking (clustering): 8 × Single-Port NVIDIA ConnectX-6 VPI 200 Gb/s InfiniBand;
- Networking (storage): 1 × Dual-Port NVIDIA ConnectX-6 VPI 200 Gb/s InfiniBand.

The cluster resources are managed by a Slurm Workload Manager (https://slurm.schedmd.com/) (accessed on 23 January 2022) [34], which is responsible for allocating the resources requested by each user for each job. The results presented in this work were obtained in a single compute node: 8×NVIDIA A100 GPUs, 256 CPUs, and 1 TB of RAM.

### 5.2. Performance Metric

Word error rate (WER) is a metric for evaluating speech-to-text models and is defined as shown in Equation (1). This metric determines the distance between transcripts by evaluating substitutions, insertions, and deletions made to a single word or word sequence in order for the transcripts to match.

$$\text{WER} = \frac{\text{Substitutions} + \text{Insertions} + \text{Deletions}}{\text{Number of Spoken Words}} \tag{1}$$

WER results range from 0 to 1 (or 0% to 100%), but these results can be higher than the upper bound if the number of additional insertions, substitutions, or deletions required for the transcripts to match is very large. WER was the selected metric for the experiments of the current work.

### 5.3. Experiments and Results

In the current work, two methods of developing models were approached, from scratch (Section 5.3.1) and transfer learning (Section 5.3.2). The from scratch method consists of an algorithm fed with data to generate a model. In transfer learning, a previously developed model acts as a starting point to develop a new model. The base model is tuned during training accordingly to new data, which are provided to the algorithm.

#### 5.3.1. Models from Scratch

The MLS dataset contains an unbalanced mix of European ($\simeq$1 h of audio recordings) and Brazilian Portuguese ($\simeq$167 h of audio recordings). Both Portuguese variants were divided into different subsets. Table 7 shows these experiments, where the European Portuguese subsets are represented by "PT" and the Brazilian Portuguese subsets by "BR". Experiments were carried out using different combinations of the variants to evaluate how data quantity affects the performance of deep models developed from scratch.

**Table 7.** Models developed from scratch using subsets of the MLS dataset.

| Train | Validation | Test | WER |
|---|---|---|---|
| PT | PT | PT | 0.9952 |
| BR | BR | BR | 0.8156 |
| BR | BR | PT | 0.8842 |
| PT + BR | PT + BR | PT | 0.8900 |
| PT + BR | PT + BR | BR | 0.8065 |

Similar from scratch experiments were performed with the SpeechDat dataset developing models with SpeechDat's train and validation subsets.

Tables 8 and 9 present the experiments carried out using the SpeechDat dataset as defined in Definitions 1 and 2, respectively.

**Table 8.** Models developed from scratch with the SpeechDat dataset as defined in Definition 1.

| Train | Validation | Test | WER |
|---|---|---|---|
| SpeechDat | SpeechDat | SpeechDat | 0.3035 |
| | | ENG | 0.9962 |
| | | PT + BR | 0.9173 |

**Table 9.** Models developed from scratch with the SpeechDat dataset as defined in Definition 2.

| Train | Validation | Test | WER |
|---|---|---|---|
| SpeechDat | SpeechDat | SpeechDat | 0.1945 |
| | | ENG | 0.9924 |
| | | PT + BR | 0.9055 |

### 5.3.2. Transfer Learning

Being that Portuguese is a low-resource language, experiments using both the Speech-Dat and MLS datasets have been developed using the transfer learning approach.

In the current experiments, various combinations of both MLS and SpeechDat were used to develop models. The 100 h "clean" training set (ENG) from the LibriSpeech dataset was used to control the performance of the transfer technique on the English domain. NVIDIA's NeMo provides a model for the English language pre-trained with a dataset with ≃3300 h of audio recordings (https://developer.nvidia.com/blog/jump-start-training-for-speech-recognition-models-with-nemo/) (accessed on 23 January 2022). This model was used as the starting point for the transfer learning experiments. Table 10 shows the results of the transfer learning experiments to compare the performance of models developed from scratch and using transfer learning.

**Table 10.** Performance of models developed using transfer learning with the MLS subsets and SpeechDat, according to Definitions 1 and 2.

| Train/Validation | Transfer | Test | WER |
|---|---|---|---|
| Pre-trained ENG | PT | PT | 1.0164 |
| | PT | BR | 0.9722 |
| | BR | PT | 0.7075 |
| | BR | BR | 0.5139 |
| | PT + BR | BR | 0.5025 |
| | PT + BR | PT + BR | 0.5083 |
| | SpeechDat 1 | SpeechDat | 0.1603 |
| | | ENG | 1.0186 |
| | | PT + BR | 0.7912 |
| | SpeechDat 2 | SpeechDat | 0.0557 |
| | | ENG | 1.006 |
| | | PT + BR | 0.7680 |

Based on the previous conclusions, further experiments were conducted by mixing SpeechDat as defined in Definition 2 and the MLS (in these experiments "MLS" represents "PT + BR", both training and testing subsets) transfer and validation subsets. The goal of these experiments is to evaluate the impact of data quality and quantity on the performance of the models.

The proportion of each subset is defined by a linear variation between 0% and 100% with a step of 25% (Table 11). This is carried out so that the number of instances remains constant across all mixes, 107,158 for training and 22,960 for validation.

**Table 11.** Audio instances from each dataset used on each training and validation mix.

| MIX ID | Transfer | | Validation | |
|---|---|---|---|---|
| | MLS | SpeechDat | MLS | SpeechDat |
| 0.00 | 0 | 107,158 | 0 | 22,960 |
| 0.25 | 9374 | 97,784 | 206 | 22,754 |
| 0.50 | 18,749 | 88,409 | 413 | 22,547 |
| 0.75 | 28,123 | 79,035 | 619 | 22,341 |
| 1.00 | 37,498 | 69,660 | 826 | 22,134 |

For each mix of the datasets, three models were developed and evaluated. Table 12 presents a summarized view of the average performance of each mix.

**Table 12.** Average performance (WER) of the models of each mix.

| MIX ID | Test Set | | | |
|---|---|---|---|---|
| | SpeechDat | ENG | MLS | SpeechDat + MLS |
| 0.00 | 0.0513 | 1.0014 | 0.7682 | 0.1912 |
| 0.25 | 0.0586 | 0.9969 | 0.5665 | 0.2155 |
| 0.50 | 0.0581 | 1.0013 | 0.3918 | 0.1221 |
| 0.75 | 0.0667 | 0.9943 | 0.3574 | 0.1225 |
| 1.00 | 0.0735 | 0.9949 | 0.3306 | 0.1231 |

The performance of the models developed in these experiments was evaluated using the previously used test sets, SpeechDat, MLS ("PT + BR"), and LibriSpeech (ENG). Additionally, a full combination of the SpeechDat and MLS test sets (SpeechDat + MLS) was also used to evaluate the model's performance. Table 13 displays the number of audio instances present in each of the mentioned test sets.

**Table 13.** Audio instances from each test set.

| Test Set | Instances |
|---|---|
| SpeechDat | 22,961 |
| ENG | 28,539 |
| MLS | 906 |
| SpeechDat + MLS | 23,867 |

*5.4. Domain Adaptation*

Within the problem at hand in this work, transfer learning should be considered not only in order to initialize the model considering a different language (English) for which there is a large volume of data, but also considering that there is limited access to data within our target (European Portuguese) and also to a variant of our target (Brazilian Portuguese). Thus, it could be questioned whether the use of instances from a dataset composed mostly of Brazilian Portuguese can be helpful for the transfer learning process between the English-optimized model and the European Portuguese target.

In order to evaluate this transfer learning domain adaptation, we proceeded with the addition of new combinations of MLS and SpeechDat datasets to add to the mixed datasets present in Table 11. These were used to train models using transfer learning. All combinations of the datasets and the corresponding WER performances can be seen in Table 14. These results correspond to the best performance attained for each (MLS, SpeechDat) mixture and cover 16 different points in the (MLS, SpeechDat) space where the number of MLS instances, $n$ is in the range $0 \leq n \leq 37,498$ and SpeechDat in the range $69,660 \leq n \leq 107,158$. The results are depicted in Figures 4 and 5.

**Table 14.** Best WER performance for all tested combinations of MLS and SpeechDat datasets.

| MLS Instances | SpeechDat Instances | WER |
|---|---|---|
| 0 | 69,660 | 0.26626 |
| | 88,409 | 0.05776 |
| | 107,158 | 0.05032 |
| 9374 | 79,035 | 0.08266 |
| | 97,784 | 0.05320 |
| 18,749 | 69,660 | 0.07130 |
| | 88,409 | 0.05602 |
| | 97,784 | 0.05408 |
| | 107,158 | 0.05398 |

**Table 14.** *Cont.*

| MLS Instances | SpeechDat Instances | WER |
|---|---|---|
| 28,123 | 79,035 | 0.06460 |
| | 97,784 | 0.05802 |
| | 107,158 | 0.05274 |
| 37,498 | 69,660 | 0.06687 |
| | 88,409 | 0.05607 |
| | 97,784 | 0.05828 |
| | 107,158 | 0.05233 |

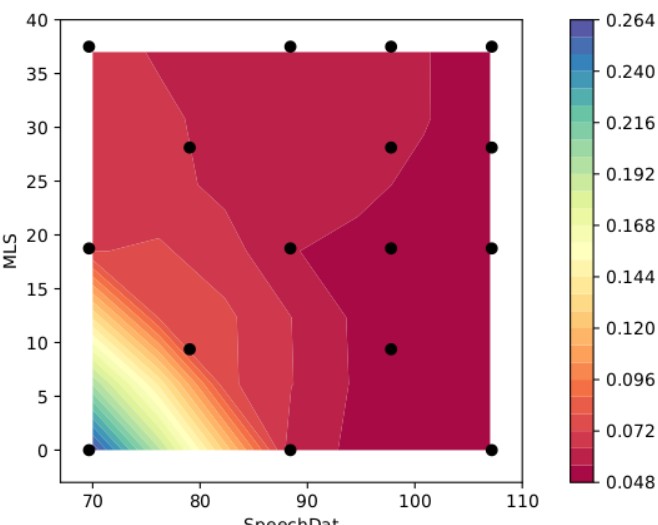

**Figure 4.** Best WER performance for all tested combinations of MLS and SpeechDat datasets (the MLS and SpeechDat axes are presented in thousands).

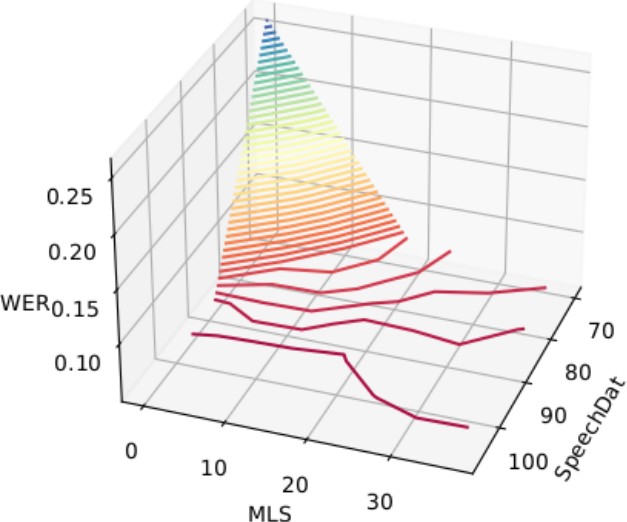

**Figure 5.** Best WER performance for all tested combinations of MLS and SpeechDat datasets (the MLS and SpeechDat axes are presented in thousands).

## 6. Discussion

The results in Table 7 reveal that the amount of data available is not enough to generate deep models with acceptable performance. The best results were achieved when training and testing the models using the largest subsets, which leads to the conclusion that larger amounts of data output better performances. Such can be seen in the second

and fifth experiments in which the models were trained and tested with the Brazilian Portuguese subsets.

Following this from scratch training approach, results in Tables 7–9 reveal that models developed using either version of SpeechDat's data (controlled environment) outperform models developed with the MLS dataset (recordings of audiobooks). Additional data pre-processing showed an increase in the model's WER performance from 0.3035 to 0.1945.

Considering the results from Section 5.3.1, it can be concluded that creating deep neural models from scratch would require more data than we have access to.

When comparing both from scratch and transfer learning approaches, some conclusions can be drawn regarding the quantity of data used during the training phase and how it affects performance.

1. The amount of data used during the training phase strongly impacts the model's performance; transfer learning improves the WER performance from 0.1945 to 0.0557.
2. It has been found that pre-trained models successfully adapt to new data by moving away from the data they were originally trained on.

Although transferring knowledge from previously developed models achieves better scores, it is observable that the transfer set still needs to have a considerable amount of data. This conclusion is supported by the following:

- Using smaller subsets, i.e., MLS subsets, the best results were achieved when using combinations with the largest subset ("BR").
- Models yielded the best results when developed with the largest subsets available, i.e., SpeechDat subsets.

These results are a reinforcement of the statement that data quality can be as important as data quantity and algorithm fine-tuning. Models developed with the SpeechDat dataset have shown an increase in performance with additional data processing even without any model fine-tuning or additional data added.

Additionally, in the mix experiments, it can be concluded that the SpeechDat dataset has the greatest impact on the performance of the models, having its performance on the SpeechDat test set slightly decreased from mix to mix. The increase in the number of instances of the MLS dataset used is reflected in the gain of performance over the MLS and SpeechDat + MLS test sets throughout the different mixes. Nonetheless, in spite of achieving better performances over these test sets, the addition of these instances does not show a negative impact on the performance over the SpeechDat and LibriSpeech (ENG) test sets.

Figures 4 and 5 represent WER as a function of (MLS, SpeechDat) and show that the worse performance (highest WER) corresponds to the minimum number of training instances (69,660). These diagrams also show that compared to point (0.69660) with WER = 0.26626, increasing the number of training instances either from the SpeechDat domain (European Portuguese) or from the MLS domain (mostly Brazilian Portuguese) generates speech-to-text systems with a clear performance increase. Comparing the performance variation per number of training instances from both domains (increasing the number of instances by 18,749) using (0.69660) as a reference, we yield the following results:

$$\Delta_{MLS} = -0.1040 \left[ \Delta WER / 10.000 \, MLS \right]$$

and

$$\Delta_{SpeechDat} = -0.1112 \left[ \Delta WER / 10.000 \, SpeechDat \right]$$

This shows similar and relevant contributions from both domains, although this contribution is more important when we increase the source with instances belonging to the same domain as the target SpeechDat (European Portuguese). Nevertheless, in the region where SpeechDat is higher than 90,000 instances, and the performance reaches values close to the maximum performance, the level curves tend to be more aligned with

the MLS axis. Thus, in this situation there is some contribution to the performance from SpeechDat but there is no significant contribution from MLS (Brazilian Portuguese).

## 7. Conclusions

The results in this work show the importance of transfer learning to the optimization of speech-to-text systems with limited data. The results demonstrate the effectiveness of the transfer learning approach to optimize a speech-to-text system with European Portuguese as the target and attain a state-of-the-art performance of WER = 0.0503, which is a clear improvement over existing ASR systems for the Portuguese language previously mentioned in Section 2.2. It was shown that the transfer learning with the SpeechDat dataset has the greatest impact on the performance of the models tested with European Portuguese SpeechDat. Varying the proportion of MLS and SpeechDat, it was clear this proportion directly influences the transfer learning process and performance for MLS and SpeechDat targets, respectively.

The use of data from the Brazilian Portuguese dataset has shown an overall increase in performance relative to situations of mid-range performance (WER = 0.2662) and little impact when in situations of near-maximum performance (WER = 0.0503). Thus, we derive the following conclusions:

- Data quantity is essential for building deep neural models, either from scratch or using transfer learning.
- The larger the amount of data from a domain used in the training or transfer process, the better the performance over the test set from the same domain.
- Data quality also plays an important role in the model's performance, since the dataset with better data quality (the SpeechDat dataset, as defined in Definition 2) yields the best performance.

**Author Contributions:** Conceptualization, investigation, methodology, writing and editing—E.M., L.C., L.R., P.Q. and P.S.; Software—E.M., L.C. and P.S. All authors have read and agreed to the published version of the manuscript.

**Funding:** This research has been partially funded by Altice Labs under contract Ref. "Optimized Portuguese Speech to Text".

**Institutional Review Board Statement:** Not applicable.

**Informed Consent Statement:** Not applicable.

**Data Availability Statement:** 3rd Party Data restrictions apply to the availability of these data. Data was obtained from Altice Labs.

**Conflicts of Interest:** The authors declare no conflict of interest.

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
