# Peer review of "Domain Adaptation Speech-to-Text for Low-Resource European Portuguese Using Deep Learning"

_futureinternet, doi:10.3390/fi15050159_

Round 1

Reviewer 1 Report

The paper presents a deep learning ASR system optimization and evaluation for the European Portuguese language. The issue addressed in this work is very important. There are many studies on ASR. However, when we are talking about low-resourced languages, the progress in this field is below expectations.

The structure of the manuscript is clear.

The main question is related to the from-scratch experiment (page 8):
If 167 hours of BP (Brazilian Portuguese) are used to train, validate and test the model, the WER is 0.8156. In commenting on the results, it is noted that the amount of data available is not enough to generate deep models with acceptable performance. While for SpeechDat (i.e. SpeechDat is used for training, validation, and testing), WER is 0.3035. SpeechDat dataset represents 186 total hours of audio recordings. That's only 19 hours more, but the result is significantly better.

The literature review section should be rewritten in light of what has been done in this area. The section "Deep learning in ASR" is general/descriptive.  It should be reviewed to see what other authors have proposed as solutions to the problem. What domain adaptation techniques are applicable to other languages?

Reviewer 2 Report

This is an interesting work in which a deep learning ASR system optimization and evaluation for European Portuguese language is described. Authors report an optimization of the model, achieving a state-of-the-art Word Error Rate (WER) of 0.0503. However, the structure of the document must be improved and more references should be included in order to validate the affirmations on this work. Also, a general revision of English grammar and typos must be carried out.

Feedback:

Abstract, Introduction and Conclusions: Authors must clearly state the age of the people who recorded the utterances of the datasets (children, young adults, adults, elderly), nativeness condition (native/non-native), background noise (yes/no), short words/sentences, general-purpose ASR or domain-specific ASR, etc.  So the hypotheses and conclusions related to this work are accurately defined and described on the paper.

The third paragraph of the Introduction must include at least one reference.

The first two paragraphs of section 2.1 needs at least one reference.

I recommend authors to include this reference on Section 2.2. ASR in the Portuguese language:

Tejedor García, C., Escudero Mancebo, D., González Ferreras, C., Cámara Arenas, E., & Cardeñoso Payo, V. (2016). TipTopTalk! Mobile application for speech training using minimal pairs and gamification. Proceedings of IberSPEECH 2016, pp. 1-8.

since they used ASR for several languages (including European and Brazilian Portuguese), and they also gathered PT speech data.

Please do not use abbreviations such as doesn't or won't, among others.

Please move the text of Definition 1 and Definition 2 to a subsection of the Method section.

Please merge Tables 5, 6 and 7 in the same table.

Please merge Tables 8 and 9 together

Pease move Table 10 and its description to the Data section

Please divide Section 5. Experiments and Results into "5. Experiments and Results and 6. Discussion". Currently, the flow of Section 5 is not clear and difficult to follow. Therefore, I highly suggest authors to just describe the results on section 5 and discuss them on a new Section 6. Also please include a brief comparison of the results with other articles/papers mentioned on Section 2.

Please report the same WER on the Abstract and Conclusions (same precision).

Round 2

Reviewer 1 Report

The article has been corrected accordingly. All review comments have been taken into account.